Symbiotic association of microalgae and plants in a deep water culture system

http://orcid.org/0000-0001-7164-0222 Özer Uyar Gülsüm Ebru 1 2 ebruozeruyar@gmail.com
Mısmıl Nurdan 1
1 Department of Horticultural Cultivation and Breeding, Institute of Natural and Applied Sciences, Kocaeli University , Kocaeli , Turkey
2 Department of Plant Protection, Faculty of Agriculture, Kocaeli University , Kocaeli , Turkey
Dąbrowski Piotr
Electronic publication date: 2022 Dec 7
Publication date: 2022
Volume: 10
Electronic Location ID: e14536
Received 2022 Sep 11; Accepted 2022 Nov 17
Copyright: © 2022 Özer Uyar and Mısmıl
Copyright year: 2022
Copyright holder: Özer Uyar and Mısmıl
License: This is an open access article distributed under the terms of the Creative Commons Attribution License, which permits unrestricted use, distribution, reproduction and adaptation in any medium and for any purpose provided that it is properly attributed. For attribution, the original author(s), title, publication source (PeerJ) and either DOI or URL of the article must be cited.
License URL: https://creativecommons.org/licenses/by/4.0/

Keywords: Chlorella vulgaris, Hydroponic systems, Deep water culture, Mint, Microalgae

Funding: Kocaeli University Scientific Research Projects Coordination Unit 2018/055 This work was supported by the Kocaeli University Scientific Research Projects Coordination Unit (Project Number: 2018/055). The funders had no role in study design, data collection and analysis, decision to publish, or preparation of the manuscript.

==============================
In this study, microalgae culture (Chlorella vulgaris) and mint seedlings (Mentha spp.) were combined in a hydroponic system to improve plant growth. Mint seedlings were grown both in microalgae-containing and in microalgae-free trial groups, and both groups were subjected to aerated and non-aerated conditions to show the effect of aeration and microalgae co-cultivation on the mint weight and height. The plant quality was also determined with color measurements of the mint leaves. The increase in the weight of the plants was the highest in microalgae-containing and aerated group (0.47 g) and the lowest in microalgae-free and non-aerated group (0.22 g). On the other hand, the variation in the plant height was not significant between the groups, the growth was lateral. The best quality mint leaves were also produced in microalgae-containing and aerated group. Our results have revealed the symbiotic life of the mint plant placed in the hydroponic system with microalgae and demonstrated improved mint growth and quality. This co-cultivation system is also potentially more environmentally friendly compared to growing microalgae and mint independently because of lower cost of aeration and mixing for microalgae cultivation, higher nutrient consumption efficiency, and reduced nutrient outflow.

Introduction

Despite the increase in the need for agricultural land as a result of the increase in the world population, the land that can be cultivated has gradually decreased (SDSN, 2013). Therefore, it is required to obtain more products from the areas suitable for agriculture. The use of chemical pesticides and fertilizers in agricultural production has increased in tandem with rising production needs. However, the chemicals used cause deformities in the soil structure and a decrease in the number of microorganisms in the soil (Rahman, Hamid & Nadarajah, 2021). While the extensive use of chemicals in agricultural fields improved production, sustaining this high soil productivity caused disruption and endangerment of the natural balance (Özdemir, Sukatar & Öztekin, 2016). For these reasons, many countries have begun to switch from conventional agricultural practices to environmentally friendly production techniques (Barbosa et al., 2015). Soilless agriculture experiments were used firstly for commercial purposes in the United States in the 1920s, and in Europe, they developed rapidly after 1965 in line with the development of technology (EI-Kazzaz & EI-Kazzaz, 2017). Soilless farming systems have been performed in two ways: hydroponics and substrate culture. With a general definition, the hydroponic system is a method of plant growing in stagnant or flowing mineral nutrient solutions without using soil. Plants get the nutrients and minerals they need from solutions given to their growing environment instead of soil (Lee & Lee, 2015).

Microalgae are located at the lowest level of the food chain, they are food of fish and other aquatic organisms, and constitute approximately one third of the plant biomass in the world (Maizatul et al., 2017). Microalgae are the most important primary producers of the ecological order and are the most valuable ecological group among organisms. In particular, recent biotechnological studies on microalgae are aimed at increasing the use of microalgae in areas such as industry, medicine, agriculture, food, feed, environment, and cosmetics (Chu, 2012; Khan, Shin & Kim, 2018). The limited fertilizers used in agriculture led to the search for new production materials, and studies are carried out to use biofertilizers in greenhouses and agricultural production areas. When biofertilizers are applied to the plant, seed, and soil surface, the living microorganisms colonize the plant and ensure the delivery of the essential nutrients it needs to the host plant (Özdemir, Sukatar & Öztekin, 2016). Chlorella vulgaris, one of the microalgae species that have very high usability in biotechnological applications, has been widely commercialized and developed in this field (José de Andrade & Maria de Andrade, 2017). This alga, which is a member of Chlorophyta, has a protein content of 43–58%. In this respect, C. vulgaris is seen as an alternative protein source, and therefore its demand is increasing in many countries (Safi et al., 2014).

Microalgae survive in hydroponic systems. It is widely assumed that microalgae will have an impact on water quality parameters such as pH, dissolved oxygen (DO), nutrient consumption in the water, and may compete with the target vegetables. Microalgae can even act as a buffer, preventing chemical sanitization with bleach treatment from being effective. As a result, algal levels in hydroponic systems must be kept to a minimum (Huo et al., 2020).

Microalgae, on the other hand, may be beneficial to plant growth in some ways. The oxygen produced by microalgal photosynthesis prevents anaerobiosis in the crop’s root system and inhibits sulfide injury in plants that are susceptible to sulfate reduction. Crop root respiration and exudation are carbon sources that boost microalgal biomass (Barone et al., 2019). A group of researchers investigated potential hydroponic systems for cultivating tomato varieties in collaboration with C. infusionum (Zhang, Wang & Zhou, 2017). According to the results, microalgal photosynthesis increased crop biomass. Also, several studies on growth-promoting substances released by microalgae in plant cultivation systems have been published in recent years (Chiaiese et al., 2018; Lu & Xu, 2015; Stirk et al., 2013). The other way to affect the plant growth is microalgal synthesis of a variety of organic non-microbial substances like phytohormones and protein hydrolysates that are active on plants (Chiaiese et al., 2018). Microalgae have been shown to secrete plant growth hormones such as auxin, cytokine, gibberellin, abscisic acid and ethylene to the growth environment. Also, using microalgae cultures (C. vulgaris) as biofertilizers in hydroponics would not interfere with lettuce germination and development. Allelochemical extracts of C. vulgaris and Scenedesmus quadricauda were discovered to be promising biostimulants in the early stages of plant growth in sugar beets by Barone et al. (2019).

Additionally, microalgae are noted for their high lipid content, which includes enriched unsaturated fatty acids and phycotoxin, which are not found in vegetables (Sathasivam et al., 2019; Zhou et al., 2011).

Moreover, antifungal and antibacterial agents are produced by algae, which helps the plant growth indirectly.

Mentha spp. are defined as mint as a general name. Mint is a medical plant and has been used in almost every field since ancient times, is from the Lamiaceae family (Can & Katar, 2020). It is originated from Central Europe and Asia; it is represented with 25 species and many hybrid species in the world. Mint is used as an aromatic spice that adds flavor to dishes and its fresh shoots and leaves are also preferred to be used in salads in Mediterranean countries. Also, peppermint oil obtained from the mint plant has wide use in the medicine, food, and cosmetics industry, and this plant is the richest natural source of menthol (Büyükbayraktar, 2014; Riachi & De Maria, 2015).

The present study was designed to analyze the effects of microalgae co-cultivation and aeration on the hydroponic plant growing system in order to improve plant growth and quality. In this context, mint seedlings (Mentha spp.) were grown in microalgae-containing (Chlorella vulgaris) and in microalgae-free hydroponic systems, under aerated and non-aerated conditions. Throughout the study, optical density of the microalgae culture, weight and height of the plants, color measurements of the mint leaves, dissolved oxygen levels were measured and analyzed. To the best of our knowledge, there is no research published on the microalgae and mint co-cultivation. Thus, the results obtained throughout this study bring a novel approach to deep water culture systems.

Materials and Methods

Microalgae and plant

In this study C. vulgaris was used as microalgae and mint (Mentha spicata) as the plant. C. vulgaris culture was supplied by the Biotechnology Laboratory, Department of Chemical Engineering, Kocaeli University, Turkey. The nutrient media used was BG11 for both microalgae and plants (Table 1). The reason for choosing BG11 was that it is suitable for the growth of C. vulgaris (B. Uyar, 2018, personal communication; Vishwakarma, Dhar & Saxena, 2019).

Table 1 Chemical composition of the hydroponic solution, prepared in distilled water; final pH was adapted as 6.8.

Components	Concentration (g L−1)	
NaNO3	1.500	
KH2PO4	0.040	
MgSO4.7H2O	0.075	
CaCl2.2H2O	0.036	
H3BO3	0.003	
Na2CO3	0.020	
C6H5FeO7.5H2O	0.006	
C6H8O7	0.006	
Vitamine mix	0.020	
Agar§	15.000	
Note:

§ Agar was added to make solid culture for preservation, propagation and preparation of pre-culture of C. vulgaris.

The stock culture of C. vulgaris was grown at the same conditions as the co-cultivation experiments in terms of nutrient media, light, temperature, and aeration. The only difference was the working volume that was 500 mL. The cell concentration was determined spectrophotometrically at 680 nm using Shimadzu UVmini-1240 Spectrophotometer (Salgueiro et al., 2016). The inoculation was made to the co-cultivation experiment cultures when the OD680 value of the stock culture reached to 2.00. The initial inoculum amount was determined to allow initial OD680 value of 0.3 to 0.4 in experimental culture. An optical density of 1.0 at 680 nm was found to correspond to a biomass concentration of 210 mgdw L−1 for this microalgae strain. While plotting the growth curve of C. vulgaris, the measured absorbance values were converted to mgdw L−1 using this data.

Hydroponic system

The deep water culture technique (DWC) was used for mints as a hydroponic system. Because the oxygen supply can be controlled in the DWC system, it is possible to determine the effects of the root formation on the microalgae growth. Moreover, it has low maintenance cost and monitoring time (Saaid et al., 2013). The DWC was prepared as 100 mL glass beakers containing 80 mL of BG11 medium as shown in Fig. 1. Aeration was supplied to the cultures with an air pump at a rate of 10 Lmin−1. Aeration provides well mixing of nutrients, homogenization of microalgae in the culture, and prevents anaerobiosis. The samples were taken from the cultures for pH, electrical conductivity (EC), dissolved oxygen (DO) and OD680 measurements. The amount of DO in the DWC was determined at the beginning and at the end of the experiments using a DO meter. pH and EC measurements were made regularly from the beginning of the experiments. pH and EC measurements were made regularly during the runs: the pH was stable between six and seven and the average EC values were measured to be ≈2.20. The measurements of temperature, pH, EC, and DO values in the samples were carried out with AZ 84051 Combo Water Quality Measuring Instrument. During the experiments, parafilm was applied to the mouthparts of the beakers to surround the beakers as seen in Fig. 1 to minimize the evaporation of the nutrient solution and prevent contamination of microalgae to microalgae-free beakers.

Figure 1 The photos of DWC system for mint production.

(A) Microalgae-free beaker. (B) Microalgae-containing beaker.

The experiments were conducted at ambient temperature (23 ± 2 °C), and under constant light intensity (mean daily photosynthetically active radiation of 29.5 μmol m−2 s−1, Lightmeter: Extech Instruments) during the entire culture period. The experiments lasted an average of 15 days. The cultures were illuminated with a 12 h photoperiod. Lighting was provided from the bottom, top and sides of the experiment cabin to provide a uniform illumination (Fig. 2).

Figure 2 DWC system set-up.

(A) Schematic diagram of the set-up, showing the lighting with yellow lines going around and bottom side of the cabin, and top lighting. (B) The photo of the experiment cabins (C) The photo showing overview of experimental design.

Analytical methods

The mint seedlings grown in pots were harvested and screened to be equal in respect to the height and the number of leaves. Selected seedlings were used in DWC experiments. The growth of mint seedlings during the experiments was evaluated in terms of increase in fresh weight, root and stem length, and leaf color. The comparisons were done between the seedlings at the beginning and at the end of the experiments. The economic yield of the mint seedlings was determined by weighting the fresh weight of them at harvest (at the end of the experiments). The weight measurements were made on the precision balance and expressed in grams (g). Root and stem length measurements were made using a digital caliper and expressed in centimeters (cm).

Color measurements of leaves (L*, a* and b* values) were made using a chromameter CR-400 (Konica Minolta Inc. Osaka, Japan) with illuminant D65 with 8 mm aperture. CIE Lab (L*, a*, b*) color identification system was used. In this color definition model, color is expressed in three dimensions (numbers):

L*: Brightness of the color (0: Black, 100: White)

a*: Foliage Redness (−60: Green, +60: Red)

b*: Blue Yellowness (−60: Blue, +60: Yellow)

After the colorimeter device was calibrated, L*, a*, and b* values were read. The instrument was calibrated with a white reference tile (L* = 97.52, a* = −5.06, b* = 3.57) prior to measurements. Three parallel samples were taken from the leaves of the samples in each experimental group, and the results were compared within the experimental groups.

Hue angle value was calculated using the following equation (Kasım, Kasım & Özer Uyar, 2015):

H=arctan(b∗/a∗)

Konica-Minolta SPAD 502 chlorophyll meter device was used for chlorophyll measurement in the study. The values obtained with the chlorophyll meter were SPAD (SPAD: Soil—Plant Analyzes Development) value.

Experimental design

The experiments were carried out in five groups with three replications and three times, each mint seedling was enumerated, and each trial was studied with a total of 12 mint seedlings. The seedlings were grouped as shown in Fig. 3.

Figure 3 The grouping and enumeration of mint seedlings at the beginning of the experiments.

G1, Group 1; G2, Group 2; G3, Group 3; G4, Group 4.

The groups planned for the experiment and the treatments applied to these groups are shown in Table 2.

Table 2 The experimental design.

Groups	Number of the seedlings	Treatments	Abbreviations	
1	1	DWC+Microalgae+Aeration	HCM1	
2	
3	
2	4	DWC+Microalgae	HCM2	
5	
6	
3	7	DWC+Aeration	HCM3	
8	
9	
4	10	DWC	HCM4	
11	
12	
5	–	Microalgae+Aeration	MAC	
Note:

DWC, Deep Water Culture; HCM, Hydroponic Culture for Mint; MAC, Microalgae culture. The notations of the treatments mentioned above have been utilized for the rest of this manuscript.

Statistical analysis

The mean values and standard deviations (SD) were calculated using Microsoft Excel™. Statistical analyses were done with one-way ANOVA in SPSS (Tukey’s HSD test) statistical package (version 25.0; SPSS, Inc., Chicago, IL, USA) to obtain critical differences at a probability level of 0.05.

Results and discussion

Growth of C. vulgaris

In pre-cultures, C. vulgaris entered the logarithmic phase 72 h after the inoculation, which is consistent to the growth curve of C. vulgaris obtained by Sakarika & Kornaros (2016) under the same conditions (Sakarika & Kornaros, 2016). The growth curves of the microalgae inoculated from the pre-culture into the beakers in Groups 1, 2, and 5 were shown in Fig. 4. The growth rate in MAC was 0.200 day−1 and the growth rates were 0.070 and 0.065 day−1 in HCM1 and HCM2 respectively. The algal growth rate in MAC was greater than the growth rates in HCM1 and HCM2 as expected, since neither nutrient nor oxygen was shared. There is a comparable study which supports this finding, the microalgal growth rate in monocultures was higher than the growth rate in co-cultures (Supraja, Behera & Balasubramanian, 2020). Since the growth curves were not very different in HCM1 (aerated) and HCM2 (non-aerated), it can be deduced that aeration is not necessary for microalgae growth in this symbiotic associated system. Aeration system (pipes, pumps, filters, energy) creates a significant cost in both microalgae cultures and hydroponic cultures (Zhang, Wang & Zhou, 2017). In this respect, it can be concluded that co-culturing strategy eliminates aeration requirement and thus provides a cost-effective and environmentally-friendly system that can be used in microalgae cultivation.

Figure 4 Growth curves of C. vulgaris for different treatments.

■ HCM1, DWC+Microalgae+Aeration; ● HCM2, DWC+Microalgae; ♦ MAC, Microalgae+Aeration.

Physical and morphological analysis of plants

The growing medium is a significant source of nutrients for plant development; hence, plant growth can be affected by the growing media. The growth of the mint was analyzed in terms of changes in total weight and shoot length. Additionally, the color and chlorophyll levels of the mint leaves were assessed across all groups.

Weight measurement of the seedlings and DO contents in the nutrient solutions

The mint seedlings were weighed at the beginning and at the end of the experiments and the weight increases were calculated, the results obtained are given in Fig. 5. The results here show both the effect of aeration and the effect of microalgae on the weight increase.

Figure 5 The change in the weights of the mint seedlings.

HCM1, DWC+Microalgae+Aeration; HCM2, DWC+Microalgae; HCM3, DWC+Aeration; HCM4, DWC, data are means ± SD (n = 9).

Aerated (HCM1 and HCM3) and non-aerated (HCM2 and HCM4) groups were evaluated to understand the effect of microalgae on plant weight.

In aerated groups, the average weight gain of mint seedlings was 0.47 g in HCM1 where microalgae were co-cultivated, while the average weight gain was 0.29 g in HCM3 which was microalgae-free. The difference between these two groups was found to be statistically significant (P < 0.05) which confirms the positive effect of microalgae on plant weight in aerated groups.

In non-aerated groups, the average weight gain was 0.30 g in microalgae-containing group (HCM2), while the average weight gain was 0.22 g in the algae-free group (HCM4). The difference between these two groups was statistically significant (P < 0.05).

The presence of microalgae during co-cultivation enhances root respiration and nutrient absorption, resulting in increased plant development (Barone et al., 2019). The secretion of secondary metabolites and other allelochemicals such as phytohormones by microalgae is another potential way for increasing plant weight (Bharti et al., 2019).

Microalgae (HCM1 and HCM2) and microalgae-free (HCM3 and HCM4) groups were evaluated to understand the effect of aeration on plant weight. The effect of aeration on weight gain was found to be statistically significant in all groups (P < 0.05).

These results suggest that aeration and microalgae have considerable synergistic effect on plant weight gain. The increase in the weight of the plants was the highest in microalgae-containing and aerated group (HCM1) and the lowest in microalgae-free and non-aerated group (HCM4).

The variations in weights or yield amongst treatments may be related also to the differences in the DO levels of the nutrient solutions. When the DO values were examined, it was seen that the DO amounts at the beginning of the experiments were higher than the DO amounts at the end of the experiment for all groups (Fig. 6). Similarly, it was shown that with an increase in culture time, there was an exponential decline in the dissolved oxygen content of the cotton hydroponic culture solution (Fu et al., 2022).

Figure 6 The percentage of DO in the DWC at the beginning and the end of the experiments.

HCM1, DWC+Microalgae+Aeration; HCM2, DWC+Microalgae; HCM3, DWC+Aeration; HCM4, DWC.

In microalgae-containing groups, the reductions of DO were 21.2% and 17.4%, in aerated (HCM1) and non-aerated (HCM2) beakers, respectively. Since HCM1 has aeration, the root system of the plant is more developed than HCM2 (Figs. 7A and 7B) which may explain the difference.

Figure 7 The root systems of the mint seedlings at the end of the experiments.

(A) HCM1, DWC+Microalgae+Aeration; (B) HCM2, DWC+Microalgae; (C) HCM3, DWC+Aeration; (D) HCM4, DWC.

In microalgae-free cultures, the reductions of DO were 23.9% and 21.1%, in aerated (HCM3) and non-aerated (HCM4) groups, respectively. Again, the reason for this difference could be the aeration which results in developing more roots in HCM3 than HCM4 (Figs. 7C and 7D). Another study showed the positive relationship between root respiration rate in the plant and the amount of DO in the nutrient solution (Ma et al., 2022). These results are also similar to the study of Zhang, Wang & Zhou (2017) in which tomato plants were used, they showed that both plant and microalgae growth were higher in the oxygen supplied groups.

Height measurement of the seedlings

Growth of mint seedlings was also analyzed in term of the changes in height. Figure 8 shows the difference in the average height measurement of the seedlings taken at the beginning and at the end of the experiments. In the height measurements at the end of the experiment, the roots of the seedlings were also included in the total length.

Figure 8 The difference in the heights of the mint seedling between at the beginning and at the end of the experiments.

HCM1, DWC+Microalgae+Aeration; HCM2, DWC+Microalgae; HCM3, DWC+Aeration; HCM4, DWC, data are means ± SD (n = 9).

Aerated (HCM1 and HCM3) and non-aerated (HCM2 and HCM4) groups were evaluated to understand the effect of microalgae on plant height.

In aerated groups, the height of all the plants co-cultivated with the microalgae was more compared to the heights obtained in microalgae-free groups. The average height gain was 4.71 cm in the algae-containing group, while the average height gain was 4.18 cm in the microalgae-free group. In non-aerated groups, the average height gain was 3.98 cm in microalgae-containing group (HCM2), while the average height gain was 3.96 cm in the microalgae-free group (HCM4). However, the effect of microalgae on height was found to be statistically insignificant in both aerated and non-aerated groups (P > 0.05). It was observed that the increase in the height of the plants growing in HCM1 was the highest among all groups, which was also the case with the weight measurements.

Cortés-Jiménez et al. (2014) discovered that the branch length of all plants co-cultivated with three microalgal concentrations was greater than the control, which had no microalgae. Barone et al. (2019) showed an increase in tomato plant shoot development when co-cultivated with Scenedesmus quadricauda. Huo et al. (2020) also observed a greater shoot growth rate for purple kohlrabi with microalgae (0.80 cm/d) compared to 0.70 cm/d in a hydroponic unit absent of algal biomass. Cortés-Jiménez et al. (2014) found that inoculating tomato seedlings with C. vulgaris in a hydroponic nutrient solution increased stem length. The presence of a microalgal population during co-cultivation promotes root respiration and nutrient absorption, resulting in higher nutrient assimilation and better plant shoot development (Barone et al., 2019).

Microalgae have been shown to synthesize plant growth hormones (Romanenko, Kosakovskaya & Romanenko, 2015; Romanenko, Kosakovskaya & Romanenko, 2016). For example, Stirk et al. (2013) showed that microalgae generate cytokinin and auxin hormone. In another study, it was determined that microalgae can produce indole acetic acid (Stirk et al., 2014). With this study, it has been shown that it supports the formation of more roots and new seedlings in plants in microalgae environments in experiments with hormones produced by microalgae.

Microalgae-containing (HCM1 and HCM2) and microalgae-free (HCM3 and HCM4) groups were evaluated to understand the effect of aeration on plant height.

In both microalgae-containing and microalgae-free groups, the height of the aerated plants was more compared to the heights obtained in non-aerated groups. However, the effect of aeration on height was found to be statistically insignificant in microalgae and microalgae-free groups (P > 0.05).

Zheng, Wang & Dixon (2007) showed in their study that insufficient oxygen in plant root zones can greatly reduce plant productivity. However little is known about the response of plants to treatments with high concentrations of oxygen in the root zones of plants. Zheng, Wang & Dixon (2007) conducted experiments on tomato plants containing O2 for 4 weeks. From the second week onwards, the tomato plant had dwarf roots and a thicker and faster root growth than the plant treated at low O2 concentration. In the same direction in this study, faster roots and lateral rootlets were formed (Fig. 7) in the HCM1 and HCM3 groups that were aerated, and the plant height elongation was higher than in the non-aerated groups. In addition, it was observed that the formation of new shoots and leaves was higher in plants in these groups. Therefore, as the O2 concentration increased, the root respiratory tract also increased linearly, and it was observed that this situation had a positive effect on plant growth.

Color and chlorophyll measurements

The L*, a*, b* values obtained from the measurements made from the leaves of mints at the end of the experiment are given in Table 3. At the beginning of the experiment, the following values were determined as: L* 39.03 ± 0.57; a* −15.77 ± 0.52; b* 23.67 ± 0.57 and the Hue 117.41 in the sample selected to represent all plants.

Table 3 L*, a*, b* and Hue values of mint leaves before and after treatments. Means with different letters indicate significant differences according to the Tukey’s HSD test (P < 0.05) (n = 9).

	L*	a*	b*	Hue value	
Initial	39.03 ± 0.57c	−15.77 ± 0.52b	23.67 ± 0.57b	117.41	
HCM1	40.88 ± 1.74b	−14.95 ± 0.27a	21.67 ± 1.08c	118.45	
HCM2	42.62 ± 1.18a	−15.98 ± 0.70b	24.73 ± 1.80ab	116.52	
HCM3	42.28 ± 0.64a	−16.25 ± 0.68bc	24.94 ± 0.70ab	116.76	
HCM4	42.93 ± 0.56a	−16.77 ± 0.35c	25.75 ± 1.22a	116.75	
Note:

HCM1, DWC+Microalgae+Aeration; HCM2, DWC+Microalgae; HCM3, DWC+Aeration; HCM4, DWC.

In the experimental groups, the L* value was the highest in HCM4 (42.93 ± 0.56), while the lowest L* value was observed in HCM1 (40.88 ± 1.74). The high L* value indicates brightness/whiteness. When looking at the mint plant in terms of both appearance and vitality, it is preferred that the leaves are darker green, therefore the color lightness of mint leaves is not a desirable condition in terms of quality, the groups with low L* value are. Therefore it can be said that microalgae-containing group with aeration has the best quality.

Similarly, negative a* values in the plant indicate that the leaves are closer to green. Compared to the microalgae-free groups, it was observed that the a* values in the microalgae-containing groups were lower. It was seen that the lowest a* value was in HCM4 with −16.77, and the HCM3 group with a relatively −16.25 value gave a close measurement value. Since the highest a* value was in HCM1, it can be concluded that microalgae-containing group with aeration is closer to green.

It was observed that the b* value, which expresses jaundice, was highest in HCM4. According to b* values, mints in HCM1 are better in terms of quality.

Hue values decreased in other experimental groups except HCM1. The hue angle value, which was 117.41 at the beginning of the trial, increased to 118.45 in HCM1 in the groups measured at the end of the trial, while it decreased to 116.52, 116.76, 116.75 in HCM2, HCM3, and HCM4 groups, respectively. The hue value is the value that determines the quality and name of the color in general. It is frequently used in the evaluation of color parameters in green vegetables, fruits and meats (Pathare, Opara & Al-Said, 2013). A higher hue value means less yellowness. As the Hue value increases, the leaves of the mints become greener.

At the beginning of the experiment, the SPAD value obtained by chlorophyll meter was 32.6 in the sample selected representing all plants, and at the end of the experiment, the SPAD values measured were 38.53 in HCM1, 38.27 in HCM2, 35.50 in HCM3 and 37.73 in HCM4. The amount of chlorophyll in the plant has been associated with various stress factors (Agathokleous, Feng & Peñuelas, 2020). Considering the results obtained in the study, the amount of chlorophyll in systems containing microalgae increased more than those without microalgae. Accordingly, it can be concluded that microalgae do not create stress on plant growth.

During the studies, no contamination occurred in the experimental groups, and no bacterial formation was observed. It is an indication that microalgae does not allow contaminations to occur. Pradhan et al. (2021) found that C. vulgaris had antibacterial activities in their study. In other studies, it has been found that extracts obtained from microalgae have antiviral, antifungal, antibacterial, antitumoral and anti-inflammatory effects (Lauritano et al., 2016; Liu et al., 2021; Sathasivam et al., 2019; Xia et al., 2021).

Conclusions

In the current study, the co-cultivation of microalgae (C. vulgaris) and mint plant in a hydroponic system was evaluated and the following conclusions were drawn:

The growth rate in microalgae culture was much higher than the growth rates in co-cultures since neither nutrient nor oxygen was shared.

Aeration and microalgae have synergistic effect on plant weight gain. The increase in the weight of the plants was the highest in microalgae-containing and aerated group and the lowest in microalgae-free and non-aerated group. The variation on weight may be related to the differences in the DO levels of the nutrient solutions.

The effect of microalgae and aeration on mint seedlings height was found to be statistically insignificant in all cases. However there were lateral growth by means of the formation of new shoots and leaves, which resulted in the significant increase in weight.

Color measurements of the mint leaves showed that microalgae-containing group with aeration has the best quality leaves.

Chlorophyll measurements showed that microalgae do not create stress on plant growth.

During the studies, no contamination occurred in the experimental groups, and no bacterial formation was observed. This is an indication that microalgae does not allow contaminations to occur.

Supplemental Information

Supplemental Information 1 Growth curves of C. vulgaris.

There were three replicates in each application and the experiments were repeated three times. For this reason, the average of nine data was taken. The Figure 4 shows the growth curves of C. vulgaris for different treatments (■ HCM1: DWC+Microalgae+Aeration; ● HCM2: DWC+Microalgae; ♦ MAC: Microalgae+Aeration)

Click here for additional data file.

Supplemental Information 2 Weight of seedlings.

There were three replicates in each application and the experiments were repeated three times. For this reason, the average of nine data was taken. In the table, the first column shows the initial values taken in the beginning of the experiments, and the other column shows the values taken at the end of the experiments. The Figure 5 shows the change in the weights of the mint seedlings (HCM1: DWC+Microalgae + Aeration; HCM2: DWC+Microalgae; HCM3: DWC+Aeration; HCM4: DWC), data are means ± SD (n= 9)

Click here for additional data file.

Supplemental Information 3 Amount of dissolved oxygen.

There were three replicates in each application and the experiments were repeated four times due to not applicable data. For this reason, the average of twelve or nine data was taken. In the table, the first column shows the initial values taken in the beginning of the experiments, and the other column shows the values taken at the end of the experiments. The Figure 6 shows the percentage of DO in the DWC at the beginning and the end of the experiments (HCM1: DWC+Microalgae + Aeration; HCM2: DWC+Microalgae; HCM3: DWC+Aeration; HCM4: DWC)

Click here for additional data file.

Supplemental Information 4 Height of the seedlings.

There were three replicates in each application and the experiments were repeated three times. For this reason, the average of nine data was taken. In the table, the first column shows the initial values taken in the beginning of the experiments, and the other column shows the values taken at the end of the experiments. The Figure 8 shows the difference in the heights of the mint seedling between at the beginning and at the end of the experiments (HCM1: DWC+Microalgae + Aeration; HCM2: DWC+Microalgae; HCM3: DWC+Aeration; HCM4: DWC), data are means ± SD (n= 9)

Click here for additional data file.

Supplemental Information 5 Color and chlorophyll measurements.

There were three replicates in each application and the experiments were repeated three times. For this reason, the average of nine data was taken. In the first row of the table, the measurements taken from a randomly selected leaf were given as the initial value. The Table 3 shows L*, a*, b* and Hue values of mint leaves before and after treatments. Means with different letters indicate significant differences according to the Tukey’s HSD test (P<0.05) (n = 9). Chlorophyll values were given and discussed in the text.

Click here for additional data file.

Additional Information and Declarations

Competing Interests

Author Contributions

Data Availability

The authors declare that they have no competing interests.

Gülsüm Ebru Özer Uyar conceived and designed the experiments, analyzed the data, prepared figures and/or tables, authored or reviewed drafts of the article, and approved the final draft.

Nurdan Mısmıl conceived and designed the experiments, performed the experiments, analyzed the data, prepared figures and/or tables, and approved the final draft.

The following information was supplied regarding data availability:

The raw measurements are available in the Supplemental Files.

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
