# Peer review of "Symbiotic association of microalgae and plants in a deep water culture system"

_PeerJ, doi:10.7717/peerj.14536_

## Round 0.1 · original submission · Major Revisions

Dear Authors,
Two independent experts provided their opinion of your work. They both agreed that the work could be published in PeerJ, but had to be significantly improved beforehand. Please read and respond to all comments from reviewers.

With best regards,

Reviewer 1 ·

Basic reporting

The format of reference are not following the journal guidance. Some references are older than 2000. Should be change with the newer one, better if using not more than 10 year-old reference.

Experimental design

Give reference for OD method measurement (line 140-143).

No information of how long experiments were carried out, except on Figure 4. Should be mentioned at Method Section.

Validity of the findings

No abbreviation begins a paragraph (line 136).

The attachment of microalgae on roots (Fig 7A) must be considered when measuring the weight of plant. Usually, the shoot and root weight are measured separately, so that the algae attachment on root system can be statistically analysed. This may explain of why there was no difference of plant height (statistically). (discussion of weight and height measurement)

Why C. vulgaris and mint are marked ‘…’ in conclusion? (line 373)

Additional comments

There are many research about the effect of microalgae to the growth of plants, either in soil or hydroponic system. Mostly are still at laboratory scale. However, the present study could present an alternative method to carry out experiment in a clean-less contaminated-controlled design. Experiment with microalgae is highly risk with bacterial or fungal (even other microalgae) contamination.

Reviewer 2 ·

Basic reporting

Manuscript correctly written but requires corrections, extension of the literature review (especially in the discussion).

Experimental design

Interestingly described part of the experiments, enriching the subject of cultivation of plants in hydroponic and aquaponic systems.

Validity of the findings

All underlying data have been provided; they are robust, statistically sound, & controlled.

Conclusions are well stated, linked to original research question & limited to supporting results.

Additional comments

The manuscript on the cultivation of mint (Mentha spp) seedlings in a hydroponic system using microalgae (Chlorella vulgaris) brings new information to this innovative method of plant management.
The manuscript fits into the profile of the PeerJ journal and, after correction, recommends the possibility of its publication.

The authors describe hydroponic cultivation. Shouldn't it be aquaponic? Please check.

Introduction: well written, however, it proposes to verify some of the references with the possibility of replacing them with newer articles.

Research methodology: please describe in detail what apparatus for water analysis?

The description of the results requires corrections, eg chapter 3.2. - is it worth presenting it if there are only 2 sentences, referring rather to the conclusions?

There is no discussion with other similar studies. If the authors want to describe the results in the chapter "Results and discussion", there must be greater reference to other studies. I suggest extending this by adding a few examples.

Conclusions are spelled correctly, but I would shorten them a bit.

---

## Round 0.2 · accepted · Accept

Dear Authors,

Both reviewers agree that your work does not require any changes and can be published in present version. My congratulations!

The Section Editor suggests a minor title edit to: “Symbiotic association of microalgae and plants in a deep water culture system” (adding the word "a").

With best regards

Reviewer 1 ·

Basic reporting

no comment

Experimental design

no comment

Validity of the findings

no comment

Additional comments

I am pleased to know that the authors had accomplished all comments from the 1st review

Reviewer 2 ·

Basic reporting

No comments

Experimental design

No comments

Validity of the findings

No comments

Additional comments

No comments